# Comparative Study of Brain Size Ontogeny: Marsupials and Placental Mammals

**DOI:** 10.3390/biology11060900

**Published:** 2022-06-10

**Authors:** Carmen De Miguel, Arthur Saniotis, Agata Cieślik, Maciej Henneberg

**Affiliations:** 1Biological Anthropology and Comparative Anatomy Research Unit (BACARU), School of Biomedicine, The University of Adelaide, Adelaide 5005, Australia; karmen.demiguel@gmail.com (C.D.M.); maciej.henneberg@adelaide.edu.au (M.H.); 2Department of Anthropology, Ludwik Hirszfeld Institute of Immunology and Experimental Therapy, Polish Academy of Sciences, Weigla 12, 53-114 Wroclaw, Poland; agata.cieslik@hirszfeld.pl; 3Institute of Evolutionary Medicine, University of Zurich, CH-8057 Zurich, Switzerland

**Keywords:** brain/body allometry, ontogenesis, brain variability, neuronal units, logarithmic/logistic curves

## Abstract

**Simple Summary:**

This study examined brain/body ontogenetic growth in marsupials and compared it with placental mammals. While marsupials display morphology and cerebral organization diverse from placentals, their neocortical arrangement and cellular composition is unclear. Unfortunately, knowledge of marsupial ontogenetic brain/body size allometry is limited. Since marsupial brain structure and volume differ when compared with those of placentals, marsupials are considered to possess simple behavioural patterns. This is misleading, since even at a basic observation, Australian marsupials display many of the same mental capacities as other mammals. Consequently, the study findings support further investigation into the intellectual abilities of marsupials.

**Abstract:**

There exists a negative allometry between vertebrate brain size and body size. It has been well studied among placental mammals but less is known regarding marsupials. Consequently, this study explores brain/body ontogenetic growth in marsupials and compares it with placental mammals. Pouch young samples of 43 koalas (*Phascolarctos cinereus*), 28 possums (*Trichosurus vulpecula*), and 36 tammar wallabies (*Macropus eugenii*) preserved in a solution of 10% buffered formalin, as well as fresh juveniles and adults of 43 koalas and 40 possums, were studied. Their brain size/body size allometry was compared to that among humans, rhesus monkeys, dogs, cats, rats, guinea pigs, rabbits, wild pigs, and mice. Two patterns of allometric curves were found: a logarithmic one (marsupials, rabbits, wild pigs, and guinea pigs) and a logistic one (the rest of mammals).

## 1. Introduction

Brain growth patterns in various mammals have been examined in the scientific literature [1,2,3,4,5,6,7]. These patterns relate to the notion of allometry introduced by D’Arcy Thompson in his book *On Growth and Form* [8]. Jerison’s extensive study of static brain/body size allometry of adult vertebrates established a method for studying mammalian brains; it did not, however, include their ontogenetic development [9]. Since, however, static allometry is a result of variations in developmental (ontogenetic) processes [10], study of ontogenetic brain size/body size allometry is important. This has been recently summarised by Montgomery et al. [11], and further discussed by Packard (2019) and Tsuboi (2019) [12,13]. Marsupials, mammals who deliver their young at an early stage of foetal development due to the lack of developed placenta and nurse them in pouches on their bodies, have been less extensively studied for brain allometries than placental mammals.

In relation to brain composition, early comparative studies of mammalian brains deemed them to be similarly fashioned, especially with regard to cerebral cortex volume and neuron/glia density ratio [14,15,16]. It has been acknowledged that there exists a negative allometry of relation of the brain size to body size in ontogeny [7]. Hawkes and Finlay also note that the frequency of neurogenesis in placental mammals tends to be fixed [7]. However, there is greater variability in marsupial brain development [7,17]. 

Although marsupials exhibit considerable diversity in their morphology, behaviour and cerebral organization, their neocortical arrangement and cellular composition is not as well understood as in placental mammals [5,16]. An exception is a commentary on low neocortical neuronal density in the opossum (*Didelphis virginiana*) [15].

Due to their distinct reproductive method, marsupials allow us to study growth of the pouch young and thus to easily observe stages of growth corresponding to intrauterine stages in placental mammals [18,19,20,21]. Although earlier studies of brain growth in marsupials do not usually include placental mammals in their comparisons [22,23,24], there has been increasing interest in marsupial brain growth [5,14,21,25]. Moreover, it has been suggested that developmental studies in marsupials constitute a relevant model for biomedical research [26]. Like placental mammals, marsupials exhibit similar neocortical organization, as well as distinct connectivity in cortical areas A1, S1, S2, V1, and V2 [5].

A recent study by Todorov et al. emphasizes that reproductive strategies and maternal investments can significantly shape the size of marsupials’ brains [27]. However, knowledge of marsupial ontogenetic brain/body size allometry is limited. Consequently, this study explores brain/body ontogenetic growth in marsupials and compares it with placental mammals.

## 2. Materials and Methods

We studied the ontogeny of brain weight relative to body weight using pouch young samples of 43 koalas (*Phascolarctos cinereus*), 28 possums (*Trichosurus vulpecula*), and 36 tammar wallabies (*Macropus eugenii*) preserved in a solution of 10% buffered formalin. The whole animal was weighed to the nearest 0.01 g. The brain was extracted by dissection and also weighed to the nearest 0.01 g. Furthermore, fresh juveniles and adults of 43 koalas and 40 possums were studied and data on body weight and brain weight were recorded.

All the animals were collected under the University of Adelaide Animal Ethics Permit 5/3/96 and South Australian National Parks Permit K23749-02. The animals studied died of natural or accidental causes in the Adelaide Hills or on Kangaroo Island. All the procedures were conducted according to the University of Adelaide ethical guidelines and regulations.

The method employed to obtain the data from the koala and possum samples is described by De Miguel & Henneberg [28]. The wallaby sample of 59 juveniles and adults was drawn from the collection of Kangaroo Island tammar wallabies prepared by Margy Wright (Department of Applied & Molecular Ecology, University of Adelaide). Body weights were taken in the field by Dr. Wright to the nearest 10 g, while the values for adult wallaby brains were estimated from measurements of endocranial volume taken by filling the skull with mustard seeds and measuring its volume to the nearest millilitre by C.D.M.

Data for the rest of the species analysed in this study were taken from the literature. References are indicated next to corresponding figures. In some publications, raw data for each specimen were available, but in most cases, only averages for each age group were published. Some data were listed in tables but for some species they were extracted from published scatterplots.

To ensure comparability of data for all species, brain sizes and body sizes were all expressed as percentages of the average adult values. A number of regression curves were fitted to the data for each species. They included linear, exponential, power, polynomial, and logistic curves. The best fitting curves in each case were selected based on their coefficients of determination (r^2^).

## 3. Results

We studied the growth of brain size compared to body size, both standardized on adult (final) values. This approach provides for the comparability of animals of very different brain sizes and body sizes. Two kinds of curves fitted the data sets: they were either logarithmic or logistic. All other types of regression gave poorer fits. Therefore, two types of brain growth patterns could be discerned: “Model A” and “Model B” (Figure 1a and Figure 2a).

Model A is characterised by fast growth in early ontogeny followed by a gradual slow-down of the growth velocity continuing into adulthood, but never ceasing completely (Figure 1a). It is well described by a logarithmic curve of the general form:BRAIN = 25 + 14 ln (BODY) [%]

The logarithmic curve fits all species equally well (R^2^ = 0.94–0.98), placental mammal and marsupial alike (possum, koala, wallaby, kangaroo, guinea pig, rabbit, and pig) (Figure 1b–h). Exact values of coefficients for each species vary a little depending on each exact data set.

Model B is characterised by fast, nearly linear growth in early ontogeny, followed by a relatively sharp slow-down to reach the asymptotic stasis in adulthood. A slight decline of brain size may happen at an old age. This pattern of growth is best approximated by a logistic curve of the general form:$$\mathrm{BRAIN}=\frac{210}{1+e^{-0.15\mathrm{BODY}}}-105\;[\%]$$

Yet again, the logistic equation provided good approximation (R^2^ = 0.90–0.99) to the growth of brain vs. body size in all placental mammal species falling into this group (humans, rhesus monkeys, dogs, cats, rats, and mice) (Figure 2a–g).

## 4. Discussion

Brain growth compared to body size is similar among mammals; however, two patterns (model A and model B) may be distinguished. Model A occurs in mammals whose body size increases continuously during adult life, e.g., in marsupials [43]. Such continuous body size increase after sexual maturity occurs in pigs [44], rabbits [45,46], and cavies [47], and hence the Model A applies to them, too. Model B shows mammals, including humans, whose body size stabilizes after reaching the adulthood. In both models, brain size remains in clear relationship to body size. This study shows that the human brain is a mammalian organ that, concerning its growth in size, is in no particular way exceptional. This is evident when comparing it to the pattern of brain growth in other mammals. Human brain anatomy is very similar to that of other primate brains [48]. The findings of our study confirm Passingham’s argument that not only is the human brain growth rate within an expected mammalian variation range, but also, that mammalian brain growth rates are more similar than body growth rates [4]. It was Ramon y Cajal who noted that mammalian brains have conserved similar anatomical features in relation to connectivity [49]. Recently, a study by Halley [50] verified that the brain growth rate minimally differed in foetal neurogenesis in placental mammals. However, this was not correlated to variations in whole body or visceral organ growth rates. Thus, during prenatal development, the brain growth rate of placental mammals is noticeably conserved [50]. It has been noted that the development of the brain and body in different vertebrates follows different ontogenetic pathways [11]. In mammals, there are two ontogenetic patterns of brain growth. In the first instance, brain growth ceases before the body is fully developed. In the second instance, brain growth is relative to body growth [11]. For instance, male eastern grey kangaroos continue to grow (especially forelimb length) much longer after females have reached mature size [51,52,53,54].

In our study, Model A indicates mammals which experience continued brain and musculoskeletal growth with eventual slowing down with age. Model B includes various species in which brain growth ceases around sexual maturity. Additionally, our study promotes further examination of the correlation between musculoskeletal development and motor neuron numbers in the brains of mammals. The basic idea here is that each motor unit of skeletal muscle is represented by a cortical neuron. Thus, the larger number of motor units is represented by larger number of cortical neurons, hence the greater brain size. This idea is illustrated by a well-known model of cortical homunculus in which human hands are represented by a large cortical area because of numerous neuronal units, while feet have a smaller representation [55]. This example may be important when comparing developmental differences in the mammals featured in Models A and B in the context of the number of motor units. At this time, we may speculate that mammals in Model A show a tendency towards increasing numbers of neurons controlling their growing musculoskeletal system in motor cortices with consequent brain size growth, since these neurons obviously form more connections and require appropriate glial cell support.

The volume of the hominin braincase has tripled in the last 3 million years (from about 450 mL to currently 1350 mL) [56]. However, evolutionary hominin brain size increase matches the increase in hominin body size [57,58,59,60]. Interestingly, human brain volume during the Holocene period has decreased by approximately 10% (100–150 mL or one standard deviation) following a reduction in the human body’s robusticity [61,62].

It has been suggested that the quality of human brain functions depends more on neurohormonal and neurotransmitter regulation than on its size [63]. Behavioural differences between mammals may also result more from neurohormonal regulation than brain size [60]. The nature of the increase in adult brain size requires further investigation, especially in the areas of neuronal connectivity and structure which reveal differences between mammalian species. For instance, it has been noted that while human frontal lobes show greater connectivity in the gPFC than in the gPFC of other placental mammals, the human frontal lobes are smaller than predicted in relation to non-human primates [64]. Another recent study (analysis of the connectome of 123 brains of various mammalian species including humans) shows that brain connectivity in mammals is identical, as well as being independent from structure and volume of the mammal’s brain [49]. The study also contends that brain connectivity in all mammals follows a universal law of conservation where the transmission of informational efficiency neural networks is equal. Both models of brain size growth described here fit a number of mammals with different evolutionary histories, positions in trophic chains, geographic locations, environmental settings, and behavioural characteristics.

Due to the fact that the brain of marsupials differs in both structure and volume when compared with that of placental mammals, marsupials are considered animals of rather simple behaviour. However, even a basic observation of the Australian marsupials shows that they display many of the same mental capacities as other mammals. They orientate well in their environment and interact with other animals and with humans in adaptive ways. Koalas, whose living environment is comparable to primates, display behaviours similar to primates: thorough judgment of supports and three-dimensional structures during climbing to treetops, and vertical clinging and leaping following careful judgment of the three-dimensional environment for distances and supports (M.H. own observations). Kangaroos, wallabies, wombats, and koalas can be tamed by humans, with whom they interact cogently and make good companion animals. However, neurophysiological studies on marsupial brain structure have not been combined with behavioural studies [65], especially in their natural environment.

It seems, therefore, that the intellectual abilities of marsupials need more investigation, as this would assist scientists in improving their knowledge of marsupial brain developmental patterns throughout their ontogeny. This would also provide a more feasible comparison of brain/body ontogenetic growth with placental mammals.

## Figures and Tables

**Figure 1 biology-11-00900-f001:**
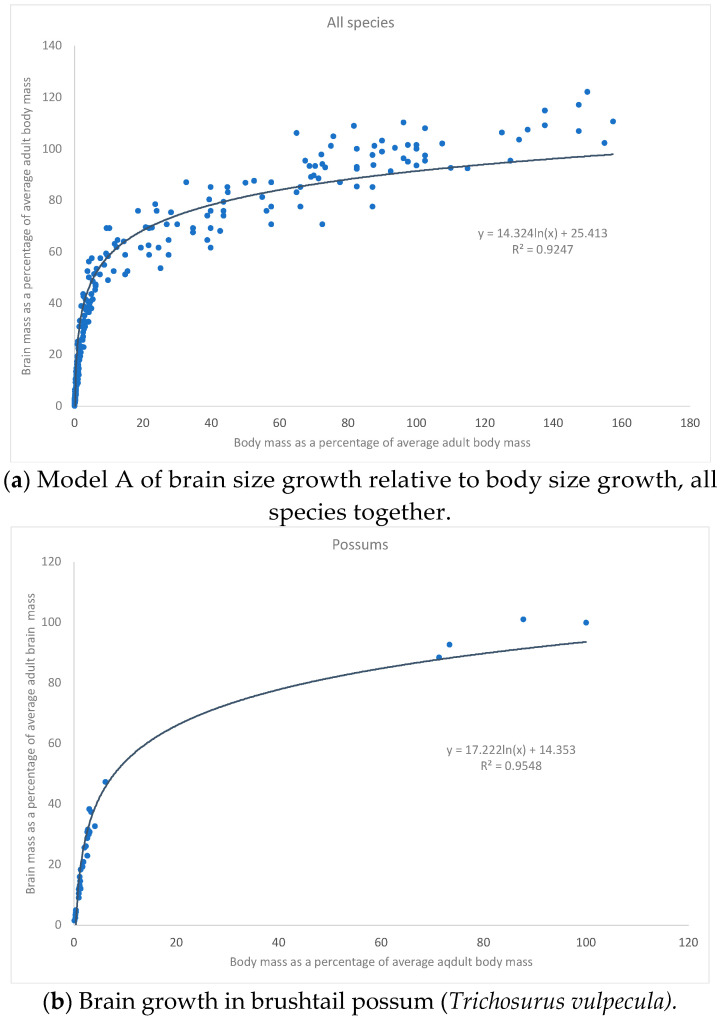
(**a**) Data from Figure 1b–h. (**b**) Data collected for this study. (**c**) Data from De Miguel and Henneberg, 1988 [28]. (**d**) Data from Janssens et al., 1997 and this study [24]. (**e**) Data from Nelson, 1988 [23]. (**f**) Data from Dobbing and Sands, 1970 [29]. (**g**) Data from Harel et al., 1972 [30]. (**h**) Data from Dickerson and Dobbing, 1967 [31].

**Figure 2 biology-11-00900-f002:**
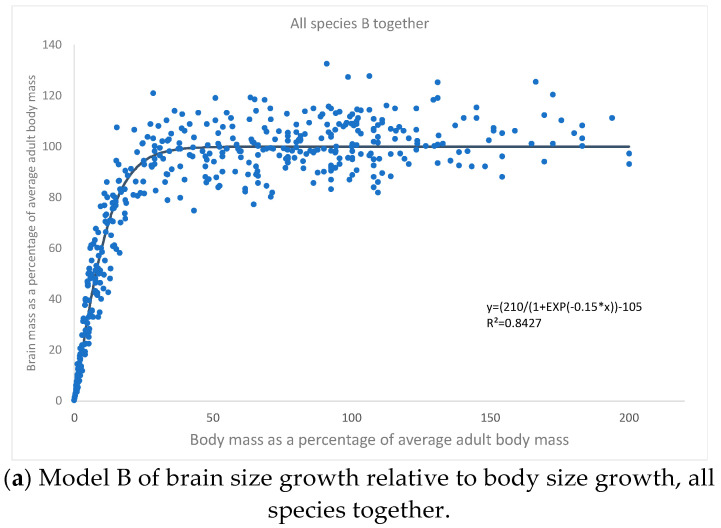
(**a**) Data from Figure 2b–h. (**b**) Data from Passingham, 1975; Harel et al., 1972 [4,30]. (**c**) Data from Count, 1947; Dobbing and Sands, 1973; Connolly, 1950; Spector, 1956; Dickerson et al.; 1982; Zilles, 1972; Kretschmann et al., 1986 [1,32,33,34,35,36,37]. (**d**) Data from Holdt et al., 1975; Falk et al., 1999 [3,38]. (**e**) Data from Agrawal et al., 1968 [39]. (**f**) Data from Agrawal et al., 1967 [40]. (**g**) Data from De Souza and Dobbing, 1971 [41]. (**h**) Data from Agrawal et al., 1968 [42].

## Data Availability

All data are available from authors on request.

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
