# Peer review of "Comparative Study of Brain Size Ontogeny: Marsupials and Placental Mammals"

_biology, 2022, doi:10.3390/biology11060900_

Round 1

Reviewer 1 Report

The authors compare the rate of brain and body size in humans and other mammals. Their aim is to challenge the idea that the human brain is unique in its development and that human abilities are not a result of brain size or morphology. They show that brain vs body growth follows one of two patterns in mammals, the difference being whether the adult ceases or continues to grow.

General Comments

The authors seem to be so focused on criticizing the idea that the human brain is unique in its growth and development that they draw conclusions from data that is insufficient to support their claims.

For example, in the introduction, the authors say that “it has been claimed that the human brain is different from other mammals especially in its fast growth in early infancy” but they do not offer any reference for such a claim. Then, in the discussion, the authors say, “the results of our study challenge the long accepted notion that the human brain size indicates the anatomical basis for Homo’s unusual abilities”

This study compares the rates of brain and body growth in mammals, including marsupials and eutherians. There is no direct comparison of mental abilities, so these results cannot directly challenge the human's uniqueness in that aspect. The results only show that there is no uniqueness in brain vs body growth rates, like Passingham (1985) had already concluded.

I suggest the authors rephrase their aim to be in line with their methodology. Besides, if their objective is to add to the discussion on the so-called uniqueness of the human brain and mental abilities, I suggest they include recent research. There is much more discussion than only that on brain growth rates - for example, Neubauer et al (2009 DOI: 10.1111/j.1469-7580.2009.01106.x) showed claimed that humans have a different ontogenetic pattern in terms of endocranial shape changes; Halley (2016 DOI:10.1159/000447254) showed that prenatal encephalization is unique to the primate order.

On their discussion on the differences between the two models of growth, the authors justify the continuous growth in Model A saying that the increased numbers of muscle motor units are associated with their continued musculoskeletal growth. I am not sure that this is the way it works. At least according to Goldring and Krubitzer (2017 DOI: 10.1016/B978-0-12-804042-3.00086-5), the representation of body areas in the brain increases in relation to their sensory-motor role, not necessarily to size. Our finger and mouth have more representation than our legs.

Specific comments

I suppose figures will be improved and not presented in the way they are currently. I suggest making it clearer what each of the axes represent. It seems that what really matters is the plot in fig 1 h and fig 2 h - so this should be placed in (a) instead of (h), or even be the only plot in the figure; It is difficult to understand what really is the original data.

Some typos: check lines 104 (“3characterised”), 136, 173, 187, 202

Is Radinsky 1979 or 1970?

line 188 - it is difficult to understand what findings concur with what...

line 252 - it has already been challenged that the frontal lobes' size is a measure of intelligence (see studies by Rilling (2006) or Semendeferi et al (2002))

Author Response

Responses to Reviewer 1

Comment 1

The authors seem to be so focused on criticizing the idea that the human brain is unique in its growth and development that they draw conclusions from data that is insufficient to support their claims.

For example, in the introduction, the authors say that “it has been claimed that the human brain is different from other mammals especially in its fast growth in early infancy” but they do not offer any reference for such a claim. Then, in the discussion, the authors say, “the results of our study challenge the long accepted notion that the human brain size indicates the anatomical basis for Homo’s unusual abilities”

This study compares the rates of brain and body growth in mammals, including marsupials and eutherians. There is no direct comparison of mental abilities, so these results cannot directly challenge the human's uniqueness in that aspect. The results only show that there is no uniqueness in brain vs body growth rates, like Passingham (1985) had already concluded.

Response: We agree with this comment. The main goal of the paper was to explore brain/body ontogenetic growth in marsupials and in comparison with placental mammals, including humans. Data on Australian marsupial brains are particularly important as they are usually difficult to obtain. In response to reviewer’s comments, the aim of our paper has changed to exploring brain/body ontogenetic growth in marsupials and compare it with placental mammals.  Most of references to humans have been removed.

Comment 2

I suggest the authors rephrase their aim to be in line with their methodology. Besides, if their objective is to add to the discussion on the so-called uniqueness of the human brain and mental abilities, I suggest they include recent research. There is much more discussion than only that on brain growth rates - for example, Neubauer et al (2009 DOI: 10.1111/j.1469-7580.2009.01106.x) showed claimed that humans have a different ontogenetic pattern in terms of endocranial shape changes; Halley (2016 DOI:10.1159/000447254) showed that prenatal encephalization is unique to the primate order.

Response: Following the suggestion, we rephrased the aim of the study, and consequently made numerous and significant changes in the manuscript: we changed the title, restructured the abstract and rephrased the Introduction section. Additionally, we added parts to the discussion in order to support our ideas and removed discussion of human uniqueness.

Comment 3

On their discussion on the differences between the two models of growth, the authors justify the continuous growth in Model A saying that the increased numbers of muscle motor units are associated with their continued musculoskeletal growth. I am not sure that this is the way it works. At least according to Goldring and Krubitzer (2017 DOI: 10.1016/B978-0-12-804042-3.00086-5), the representation of body areas in the brain increases in relation to their sensory-motor role, not necessarily to size. Our finger and mouth have more representation than our legs.

Response: We made this fragment of discussion clearer and concise.

(…) Additionally, our study promotes further examination of the correlation between musculoskeletal development and motor neurons numbers in brains of mammals. The basic idea here is that each motor unit of skeletal muscle is represented by a cortical neuron. Thus, the larger number of motor units is represented by larger number of cortical neurons, hence the greater brain size. This idea is illustrated by a well-known model of cortical homunculus in which human hands are represented by a large cortical area because of numerous neuronal units while feet have a smaller representation. This example may be important when comparing developmental differences in the mammals featured in Models A and B in the context of the motor units number. At this time we may speculate that mammals in Model A show a tendency towards increasing numbers of neurons controlling their growing musculoskeletal system  in motor cortices with consequent brain size growth since these neurons obviously form more connections and require appropriate glial cells’ support (…).

Reassuming, we put the hypothesis based on a piece of general knowledge as there is a very scarce bibliography regarding motor units and their cortical representation in marsupials.

Comment 4

I suppose figures will be improved and not presented in the way they are currently. I suggest making it clearer what each of the axes represent. It seems that what really matters is the plot in fig 1 h and fig 2 h - so this should be placed in (a) instead of (h), or even be the only plot in the figure; It is difficult to understand what really is the original data.

Response: The figures were placed in the order suggested by the Reviewer and their quality was improved

Comment 5

Some typos: check lines 104 (“3characterised”), 136, 173, 187, 202

Response: the typos were corrected

Comment 6

Is Radinsky 1979 or 1970?

Response: 1979; corrected

Comment 7

  1. line 188 - it is difficult to understand what findings concur with what...
  2. line 252 - it has already been challenged that the frontal lobes' size is a measure of intelligence (see studies by Rilling (2006) or Semendeferi et al (2002))

Response: Lines have been omitted as discussion has been altered.

Reviewer 2 Report

General Comments: There are interesting data in this study regarding brain size through development for a number of marsupials. While these data are interesting and important, the scope of the study as written is not particularly broad. Below I outline several questions and suggestions that might strengthen the manuscript.

  • This is a very descriptive paper with lots of interesting data on marsupials – arguably a challenging group to acquire data like this for. However, the paper could be strengthened with a better tie to scaling relationships across animals. For example, what do we know about how other body parts scale with body size across animals? If there was something about the brain that was special, we might expect this scaling relationship to break down with the brain, right? I think there are some clear predictions about brain size generally that could be incorporated. Along these lines, I did not find the focus on human brain growth particularly compelling as a primary narrative for the manuscript.
  • Is it typical for this journal to include detailed methods and sample sizes in the abstract?
  • Is overall brain size the most appropriate measure for the questions at hand? More discussion about how what exactly different measures of brain “size” or “complexity” can tell you about potential brain function would greatly strengthen the paper.
  • In the introduction, it might be useful to briefly describe what eutherians and marsupials are for any non-mammologists reading the article.
  • The figures did not have any axes legends
  • The discussion beging with a statement about the relationship “between brain size growth and intellectual/social complexity”, yet this study did not look at all at “intellectual/social complexity”, nor did it discuss these qualities/traits in any of the focal taxa.
  • Again, this study is relevant to the allometric growth literature, but this literature was barely mentioned. I think focusing on this framework would strengthen the manuscript and broaden its appeal.
  • The last sentence of the 1st paragraph of the discussion has no citation, yet this seems to be the basis for the paper as written.
  • On page 10 in the 2nd to last paragraph, it would be useful to provide more information about other vertebrates. For example, as written it says “in contrast to other vertebrates and mammals….”, without mention of which other vertebrates. This study is more interesting from a comparative perspective, and so knowing more about the other groups for comparison is helpful.

Author Response

Responses to Reviewer 2

Comment 1

This is a very descriptive paper with lots of interesting data on marsupials – arguably a challenging group to acquire data like this for. However, the paper could be strengthened with a better tie to scaling relationships across animals. For example, what do we know about how other body parts scale with body size across animals? If there was something about the brain that was special, we might expect this scaling relationship to break down with the brain, right? I think there are some clear predictions about brain size generally that could be incorporated. Along these lines, I did not find the focus on human brain growth particularly compelling as a primary narrative for the manuscript.

Response: Sample sizes are a piece of general data in the context of the research scope. However, the abstract as a whole has been rewritten.

Comment 2

In the introduction, it might be useful to briefly describe what eutherians and marsupials are for any non-mammologists reading the article.

Response: We entered the terms marsupials and placental mammals instead of eutherians to be more precise in this matter. The whole manuscript contains many facts regarding different aspects of marsupials anatomy and ontogeny, we also added a short paragraph about their behavioural patterns in the Discussion section. We are sure that even a non-mammologist could possibly find a lot of interesting information about these fascinating animals. We are also sure that most scientists understand what marsupials are. This is a common zoological term.

Comment 3

The figures did not have any axes legends.

Response: This was probably due to technical issues resulting from transmission of the Word file created in the Mac system Monterrey 12.2 to the Microsoft Windows system. This has been resolved now.

Comment 4

The discussion begging with a statement about the relationship “between brain size growth and intellectual/social complexity”, yet this study did not look at all at “intellectual/social complexity”, nor did it discuss these qualities/traits in any of the focal taxa.

Response: We agree with the comment, thus we changed a lot of elements of the manuscript: first of all, we changed the title,  rephrased the Introduction section, we also improved the Discussion section as well as added several new bibliographic sources.

Comment 5

Again, this study is relevant to the allometric growth literature, but this literature was barely mentioned. I think focusing on this framework would strengthen the manuscript and broaden its appeal.

Response: We have cited fundamental works (eg. Jerison 1973) relating to allometry, for example D’arcy Thompson.

Comment 6

The last sentence of the 1st paragraph of the discussion has no citation, yet this seems to be the basis for the paper as written.

Response: As a consequence of other significant changes implemented in the text, this sentence was removed.

Comment 7

On page 10 in the 2nd to last paragraph, it would be useful to provide more information about other vertebrates. For example, as written it says “in contrast to other vertebrates and mammals….”, without mention of which other vertebrates. This study is more interesting from a comparative perspective, and so knowing more about the other groups for comparison is helpful.

Response: This sentence has been omitted. Classic studies of Jerison included fish, amphibians and reptiles. These are “other vertebrates” we refer to. We do not think that in a brief discussion there is a need to name all vertebrate classes.

Reviewer 3 Report

Maybe add some key words. I like the work very much. The conclusion "the pattern of human brain growth is not different from that of other 17
mammals, both marsupials and eutherians", do not surprise me, because Homo sapiens is a mammal. But it is good that it is proved.

Author Response

Comment 1

Maybe add some key words. I like the work very much. The conclusion "the pattern of human brain growth is not different from that of other 17
mammals, both marsupials and eutherians", do not surprise me, because Homo sapiens is a mammal. But it is good that it is proved.

Response: We have added the following keywords: Brain/body allometry, ontogenesis, brain variability, neuronal units, logarithmic/logistic curves

Round 2

Reviewer 1 Report

The manuscript is now clearer. Given the lack of studies on marsupials, it is important that your study focuses on this mammal group.

Just some minor comments:

Discussion:

1st sentence: "Brain growth against the body size is similar among mammals" - maybe rewrite: "brain size against body size during growth" or "Brain growth against body growth"

Last paragraph (p.28): "It seems, therefore, that the intellectual abilities of marsupials need more investigation" - I would remove "It seems, therefore" and go direct to the point "The intellectual abilities..." and the last sentence "This would also provide" "this" refers to your study or to the future investigation of marsupial cognition?

One question: is the ability to continue growing throughout life related to their pouch development?

Please check punctuation:

Line 92 - "marsupials' brains, However, knowledge" and end of page 25 "visceral organ growth rates. Thus, during prenatal development"